# Expression of HERV Genes as Possible Biomarker and Target in Neurodegenerative Diseases

**DOI:** 10.3390/ijms20153706

**Published:** 2019-07-29

**Authors:** Antonina Dolei, Gabriele Ibba, Claudia Piu, Caterina Serra

**Affiliations:** Department of Biomedical Sciences, University of Sassari, Viale San Pietro 43B, 07100 Sassari, Italy

**Keywords:** HERV human endogenous retroviruses, HERV-Wenv, HERV-Kenv, multiple sclerosis, amyotrophic lateral sclerosis, neuroAIDS, neurodegeneration, TDP-43

## Abstract

Human endogenous retroviruses (HERVs) are genetic parasites, in-between genetics and environment. Few HERVs retain some coding capability. Sometimes, the host has the advantage of some HERV genes; conversely, HERVs may contribute to pathogenesis. The expression of HERVs depends on several factors, and is regulated epigenetically by stimuli such as inflammation, viral and microbial infections, etc. Increased expression of HERVs occurs in physiological and pathological conditions, in one or more body sites. Several diseases have been attributed to one or more HERVs, particularly neurological diseases. The key problem is to differentiate the expression of a HERV as cause or effect of a disease. To be used as a biomarker, a correlation between the expression of a certain HERV and the disease onset and/or behavior must be found. The greater challenge is to establish a pathogenic role. The criteria defining causal connections between HERVs and diseases include the development of animal models, and disease modulation in humans, by anti-HERV therapeutic antibody. So far, statistically significant correlations between HERVs and diseases have been achieved for HERV-W and multiple sclerosis; disease reproduction in transgenic animals was achieved for HERV-W and multiple sclerosis, and for HERV-K and amyotrophic lateral sclerosis. Clinical trials for both diseases are in progress.

## 1. Introduction

The human genome is, on average, composed of 8% human endogenous retroviruses (HERVs) [1], which, in some way, are entities in-between genetics and environment [2]. In fact, the HERVs are the remnants of ancient infections of our progenitors by regular exogenous retroviruses. Being transmitted through generations in our DNA, they may be considered genetic elements that could also differ among individuals. Deriving from exogenous retroviruses, and behaving for several functions as viruses, they come from the environment, and may be activated by environmental factors and infectious agents [2,3]. More likely, HERVs could be viewed as genetic parasites [4]. HERVs are highly defective, but few complete proviruses have retained the classical genome organization of retroviruses, with two long terminal repeat regions (LTR, transcriptional promoters), including the *gag* (coding for the structural matrix, capsid and nucleocapsid proteins), *pol* (coding for the reverse transcriptase, integrase and protease proteins) and *env* (coding for the envelope surface and transmembrane proteins) genes [1,3,5]. There are several nomenclature systems for classification of endogenous retroviruses [6]. We will follow the most used one, which utilizes the amino acid specificity of the tRNA that binds to the primer-binding site, to trigger the reverse transcription reaction, by adding its one-letter code as a suffix to the HERV acronym: HERV-W uses a tryptophan-specific tRNA primer, while HERV-K uses a tRNALys [7].

Over time (5–70 million years), deleted or mutated versions of retroelements have accumulated in the DNA. Some transposable elements have become inactive, others have retained mobility within the genome; variably inserting in cellular genes, and differentially within allotypes of polymorphic genes, these elements have determined inheritable, stable gene modifications [7]. Moreover, they may supply new promoters through their LTRs, create new functional exons, alternative splicing products and miRNAs, via integration and adaption events, hence contributing to DNA plasticity and evolution. The majority of HERVs are also found in old-world monkeys; the most recently integrated HERVs have polymorphisms between human populations and individuals [8]. Only the HML-2 subgroup of HERV-K is present exclusively in humans, and some insertional polymorphism between individuals was detected [9].

To survive in such consistent amounts in the genome, mutual adaptations must have occurred in the HERVs.

In some cases, the host has the advantage of some HERV genes, to serve physiological functions, as for the exaptation of the fusogenic properties of the Syncytin-1 env protein, the only intact ORF coded by a replication-incompetent HERV-W provirus of locus 7q21.2 (ERVWE1). During early pregnancy, Syncytin-1 is expressed specifically on the surface of blastocyst’s cytotrophoblasts, allowing their fusion into the multinucleated syncytiotrophoblast layer, thereby allowing embryo implantation [10].

Several possible beneficial effects have been reported [7], which include embryo implantation, gene variability, alternative splicing and polyadenylation, mother immunosuppression to protect the fetal allograft, protection against superinfection by exogenous retroviruses, involvement in development and/or differentiation, etc.

Conversely, the HERVs can also contribute to generating inherited diseases, by induction of gene instability and variability, recombination, gene disruption by insertion of HERV sequences, autoimmunity, superantigenic stimulation, production of immunosuppressive factors, cancer through interactions with oncogenes, activation/inactivation of growth controlling genes, etc. [7,11].

The expression of different HERVs varies, even within the same individual, depending on several factors. The chromatin region, where the HERV is located, must be available for transcription (DNA methylation, chromatin remodeling, etc.); this may change in relation to programs of development, to cell histotype, differentiation and proliferation, to oncogenesis and/or tumor progression, etc. [12]. When the coding potential is retained, the retroelement undergoes the same regulation of surrounding cell genes.

The above baseline predisposition of retroelements to be expressed may be regulated epigenetically by different stimuli, such as inflammation, viral and microbial infections, etc. [2,7].

Therefore, it is no wonder that increased expression of one or more HERVs has been detected in various physiological or pathological conditions, in one or more body sites. Several clinical situations have been claimed to be attributable to one or more HERVs, as reviewed elsewhere, particularly with respect to some neurological diseases, as discussed below [2,3,7,13,14,15,16].

A link between HERVs and neurodegeneration is not surprising, since regular exogenous human retroviruses also always cause some neuropathology [14]. HIV invades the central nerve system (CNS) within days of infection [17] and neuroAIDS (or HIV-associated neurocognitive disorder) is a well-recognized syndrome [18]. HTLV-I, apart from Adult T cell Leukemia/lymphoma, may cause the tropical spastic paraparesis, which is a chronic systemic immune-mediated inflammatory myeloneuropathy [19].

With respect to neuropathogenesis, the products of the *env* genes are the most studied among the gene products of the various HERVs, in view of the properties of the protein, which has a surface moiety (SU, with a putative receptor function, and, presumably, the fusogenic, superantigenic and immune-activating/pro-inflammatory properties) and a trans-membrane moiety (TM). The latter domain has an extracellular region (with a potentially immunosuppressive domain), an intra-membrane domain, and an intracytoplasmic tail (responsible for signaling) [4,20]. *In vivo*, intraperitoneal injection of purified HERV-W/MSRV into humanized SCID mice caused a T-mediated neuropathology [21], and HERV-W/Syncytin-1 caused the death of oligodendrocytes, neuro-inflammation, and neuro-behavioural deficits in transgenic mice [22].

The key problem is how to differentiate the expression of a HERV as *cause* or *effect* of a particular disease. To be used as a biomarker, a statistically significant correlation between the expression of a certain HERV and the disease onset and/or behavior must be found, by specific and validated assays. The greater challenge is to establish a role in the pathogenesis. The criteria required to establish causal connections between HERVs and disease have been discussed [4], and include the development of animal models, and attempts to modulate the disease in humans, by blocking the endogenous retroviral protein function with therapeutic antibody.

So far, a statistically significant correlation between HERV expression and a disease has been repeatedly and independently achieved only for HERV-W and multiple sclerosis (MS), while the reproduction of human diseases in transgenic animals was achieved for HERV-W and multiple sclerosis and for HERV-K and amyotrophic lateral sclerosis (ALS), as described below. The schematic structures of HERV-W and HERV-K(HML-2) elements are shown in Figure 1.

## 2. HERV-W and Multiple Sclerosis

MSRV (multiple sclerosis-associated retrovirus) was the first HERV identified [23], released by cultured leptomeningeal cells from the cerebrospinal fluid (CSF) of an MS patient. MSRV is the founder member of a new HERV family, named HERV-W after its tryptophan-specific t-RNA binding site, and its extracellular RNA has the typical retrovirus organization [20,24]. MSRV*env* and Syncytin-1 are transcribed by separate HERV-W elements, have >94% identity at RNA level, and can be discriminated only at this level, by selective primers, the specificity, sensitivity and validation of which has been reported [25], and which, at the protein level, are indistinguishable.

### 2.1. HERV-W/MSRV as a Biomarker for MS

To date, the presence of extracellular MSRV particles, and the expression of HERV-W/MSRVenv and Syncytin-1 transcripts have been detected repeatedly and independently, by several groups, in blood, brain and CSF of MS patients from different Caucasian populations, and found to be increased with respect to both healthy and pathological controls, as reviewed in [2,3,14]. Also, the HERV-Wenv protein was found increased, as detected by immunohistochemistry on brain cells and by flow cytometry on circulating B, NK and monocytes and on macrophages.

Reliable clinical predictors are crucial for identifying appropriate candidates for early or aggressive therapies, or for follow up monitoring. The ideal biomarker should predict to each individual, his/her own clinical course and sensitivity to therapy.

Over time, our group performed a series of longitudinal studies on HERV-W/MSRVenv expression in MS patients and controls, as reviewed in [2,13].

A disease frequently prodromic to MS is optic neuritis (NO, *Neuromyelitis optica*). In a ~two-year follow up of a Swedish cohort, we observed that HERV-W/MSRV positivity in the blood and CSF of patients at NO onset was significantly higher than in pathological controls. At follow-up, 11 out of 49 NO patients developed clinically definite MS. These patients represented the 33% of the MSRV(+) CSF group and 0% of the MSRV(-) group (p < 0.03), indicating that the conversion to full-blown MS in this short time interval occurred only among those patients who had already HERV-W/MSRV-positive CSF [26].

We found repeatedly, and in different cohorts, that HERV-W/MSRV positivity and load (in blood, CSF and brain samples) has a direct parallelism with MS status, years of disease, active or remission phases, and clinical stages [2,13,27,28].

In a blind ten-year follow up from the onset of MS patients with similar features, but different for HERV-W/MSRV positivity in the CSF, we found that the above CSF positivity at MS onset was related to poor prognosis. Along the course of the follow up, after three [29], six [30] and ten years [31], mean EDSS (expanded disability status scale), annual relapse rate, and therapy requirement were significantly higher in patients who had detectable HERV-W/MSRV CSF load at the onset; of note, 43% of the patients with HERV-W/MSRV in the CSF at onset progressed to secondary-progressive MS, while those with HERV-W/MSRV(-) CSF at the onset remained in relapsing-remitting MS, as shown on Figure 2. This means that, in patients starting from otherwise comparable conditions, the worst progression had signals ten years earlier in that cohort.

Therefore, we proposed that the evaluation of HERV-W/MSRV should be considered the first prognostic marker for the individual patient, to monitor disease progression.

We observed that when MS patients undergo successful therapies, the expression of HERV-W/MSRV diminishes accordingly. In a longitudinal evaluation of MS patients, undergoing efficacious therapy with interferonβ, we found that HERV-W/MSRV viremia fell rapidly below detection limits (the earliest effect was detected 48 h after the first drug shot). Interestingly, a patient, after initial clinical and virological benefits, had a HERV-W/MSRV rescue, which started three months before a strong disease progression (from EDSS 2 to EDSS 5) and therapy failure [28]. Our findings were confirmed by an independent study, which observed a significant decrease in anti-HERV-Wenv and anti-HERV-Henv antibody reactivity, in relationship to interferonβ therapy, closely linked to efficacy of therapy/low disease activity [32].

In another longitudinal study, we showed that Natalizumab (a humanized monoclonal antibody highly effective against relapsing-remitting MS) strongly reduces the expression of HERV-W/MSRV/Syncytin-1, in parallel with the clinical benefit [33]. In a series of horizontal studies, comparing MS patients with/without therapy, we found reduced expression of HERV-W/MSRV/Syncytin under therapy. This was true for Interferonβ, Natalizumab, Fingolimod, Azathioprine and Glatiramer acetate, as reviewed in [2]. The reduction of HERV-W/MSRV expression in our hands ranged from 4 to ~1 Log_10_, with respect to study entry (in longitudinal studies) or to untreated control MS patients (in horizontal studies).

All of our above data, and independent findings from other groups [32,34], strengthen our hypothesis that the evaluation of HERV-W/MSRV could be considered the first prognostic biomarker for the individual patient, to monitor disease progression and therapy outcome.

At the DNA level, increased HERV-W*pol* DNA sequences were observed in MS, by *in situ* hybridization, and proposed as a potential marker of MS [35]. By discriminatory PCR assays, we detected a mean 6-fold increase of HERV-W/MSRV*env* DNA copy number in MS patients with respect to healthy controls, but unchanged numbers of HERV-W/Syncytin-1 DNA copies [25]. A subsequent study found a statistically significant 19% increase of MSRV*env* DNA copies, influenced by gender and disease severity [36].

### 2.2. HERV-W/MSRV as Contributor to MS Pathogenesis

The aetiology of MS is unknown, but thought to be multifactorial; its immunopathogenic phenomena are triggered by environmental factors operating on a predisposing genetic background [37]. Among other effects, the inflammatory process hits and destroys oligodendrocytes, the cells that produce the myelin sheaths around axons in the brain and spinal cord.

A variety of viruses were proposed to contribute to MS risk, either by infecting the brain, or activating peripheral, cross-reactive T-cells, acting against nerve myelin [38]. In most cases, the link with MS was weak, apart from the Epstein–Barr virus (EBV) and for HERV-W/MSRV/Syncytin-1, as reviewed in [2,3,39,40,41].

EBV-infected persons have a higher risk of developing MS, especially if they have EBV in late adolescence or adulthood, when the infection is symptomatic (infectious mononucleosis). An umbrella review of meta-analyses found credible evidence of MS risk for infectious mononucleosis, anti-EBV-EBNA IgG seropositivity, and smoking, by unclear mechanisms [42], and it remains to be determined whether EBV plays a role after MS initiation [2,3,41].

We found that EBVgp350, the major envelope protein of EBV, activates HERV-W/MSRV in vitro in blood cells and in astrocytes [43], as reported also on Figure 3. HERV-Ws are not expressed by T cells; in basal conditions the most active cells are the natural killer cells; when monocytes differentiate into macrophages, the expression of HERV-W/MSRVenv and of HERV-W/Syncytin-1 increases of 6.5-fold and 3-fold, respectively, indicating that the two HERV-W elements have different regulatory patterns, as also seen under HIVtat stimulation [44]. In another study, we found that EBV activates

HERV-W/MSRV also *in vivo*, in patients with infectious mononucleosis and in healthy humans with high anti-EBNA1 IgG titers [45]. These data indicate that the two main links between EBV and MS (infectious mononucleosis and high anti-EBNA1-IgG titers) are paralleled by activation of the potentially neuropathogenic HERV-W/MSRV. Considering our data and those of the literature on MS pathogenesis, we postulated the possibility for EBV of an initial trigger of future MS, years later, and for HERV-W/MSRV of a direct role of effector of neuropathogenesis before and during MS [2,45].

In fact, all of the above data and reports indicate that, whatever the initial trigger, MS patients have a higher presence (higher DNA copy numbers) and expression (mRNA and protein) of the immunopathogenic and neuropathogenic HERV-W/MSRV, which increases with disease onset and progression, and is inhibited by efficacious therapies.

Attempts to reproduce the MS disease in animal models were done: HERV-W/MSRVenv protein alone was shown to promote experimental allergic encephalomyelitis in mice, i.e., the animal model of human MS [46], with a reversal of clinical score kinetics, if the mice were treated with an anti-HERV-Wenv GNbAC1 monoclonal antibody [47].

The GNbAC1 antibody overcame a phase 1 clinical study of healthy subjects and MS patients [47], and was tested in a phase 2b clinical trial on relapsing MS patients (CHANGE-MS Project, ClinicalTrials.gov Identifier: NCT02782858, first posted: 25 May 2016). The in-human trials of this antibody have been recently reviewed [48]. According to this review, no relevant issues with tolerability or safety have been described to date, and the treatment concept of GNbAC1 is appealing, but remains controversial, since the anticipated immunomodulatory effects were not observed in clinical or pharmacodynamic analyses of the published data on patients. However, press-released 48-week results announced a significant decrease of neurodegenerative brain atrophy in GNbAC1-treated MS patients [49]. This effect could be accounted for by the findings of *in vitro* co-culture experiments, showing that the HERV-Wenv protein, via myeloid cells, directly harms axons, and that the damage can be overcome by anti-HERV-Wenv antibody [50,51]. If the press-released positive results of the trial will be published, then all of the criteria required to establish causal connections between HERV-W and MS disease [3] will be fulfilled.

## 3. HERV-K and Amyotrophic Lateral Sclerosis

Apart from HERV-W and MS, of the dozens of other HERV families, only HERV-K was repeatedly reported to have potentially pathogenic properties, with respect to cancer [52,53,54] and to ALS (see below). The expression of HERV-K, especially of the HERV-K(HML-2) subgroup, has been associated with many cancers such as prostate cancer, melanoma, teratocarcinoma, ovarian, and germ cell tumours, as reviewed in [53]. Three mechanisms potentially involved in HERV-K oncogenesis have been described: i) the LTRs, which may enhance transcription of host genes; ii) HERV-K-encoded proteins, such as Np9 (a pivotal switch of several signaling pathways), and Rec (de-repressor of oncogenic transcription factors), both splicing products of the *env* gene; iii) the cytoplasmic tail of HERV-K(HML-2) Env, which, unique among the retroviral Env proteins tested, is oncogenic *per se*, is a strong inducer of several transcription factors associated with cellular transformation, as reviewed in [55]. This was confirmed by our finding that the disruption of the HERV-K(HML-2)*env* gene causes the downregulation of EGF-R and NF-κB, pivotal signaling pathways, central to cancer and immune responses [55].

### 3.1. Amyotrophic Lateral Sclerosis

ALS is a no-therapy neurodegenerative disease, with progressive loss of motor-neurons, muscle weakness, paresis and respiratory failure, and is 90% fatal within 5 years after onset. The etiology is unknown, but complex and multifactorial, with a probable multistep interplay of multiple genes, environmental and age-related factors (diet, athleticism, heavy metals, β-N-methylamino-L-alanine neurotoxin from Guam cycad seeds and cyanobacteria, viruses, etc.) and stochastic events [56]. The ALS cases are familial by 5%, and sporadic by 95%. There are several ALS-associated mutations: SOD-1 (superoxide dismutase), TAR DNA-binding protein 43 (TDP-43, transactive response DNA binding protein 43 kDa), C9orf72 (chromosome 9 open reading frame 72) protein, RNA-binding protein FUS/TLS (Fused in Sarcoma/Translocated in Sarcoma), etc. The multiple pathogenic mechanisms of ALS (including TDP-43 mutations) share cytoplasmic TDP-43 deposits as a common trait, which are thought to be critical in the degenerative process of motor neurons, and considered the final hallmark of ALS. The ALS prevalence is 0.4–0.6/1000 persons, but it is 10-fold higher in HIV+ persons [3]. Of note, HIV-associated ALS can be reversed by antiretroviral drugs [57].

Since 1975, retroviral markers (reverse transcriptase activity and/or virus-like particles have been independently found in brain lesions, blood and spinal fluids of ALS patients (around 45–50% of ALS cases), and, with lower frequency, in their blood-relatives, but not (or much less) in unrelated healthy and pathological controls [58,59,60,61,62]. However, it was not possible to attribute these markers to any known retrovirus [62,63,64].

### 3.2. HERV-K Expression in Vivo in ALS Patients and Controls

The expression of HERV-K has been evaluated in autopsied brain tissues of ALS patients and controls by the group of A. Nath (National Institute of Health, Bethesda, Md) [65]. HERV-Kpol expression was found in cortex motor neurons, but not in astrocytes, and the HERV-K reverse transcriptase protein strongly co-localyzed with the TDP-43 protein; it was defined as a specific pattern of expression, including a unique HERV-K 7q34 locus more frequently expressed in ALS patients than in the controls. Interestingly, this genomic region is adjacent to the 7q36.1 region, previously associated with motor neuron disease [65].

A subsequent study from the same group found HERV-Kenv expression in cortical and spinal neurons of ALS patients, but not in neurons from control healthy individuals [66]. Expression levels of HERV-K *pol*, *env*, and *gag* genes correlated with each other, suggesting that an entire HERV-K genome was activated; the env protein was detected in neurons of ALS patients, but not in the brains of healthy or pathological controls [66].

Importantly, they created transgenic mice expressing the HERV-K*env* gene. These animals developed an ALS-like progressive motor dysfunction, accompanied by selective loss of volume of the motor cortex, specific loss of upper and lower motor neurons, and other damages. Injury to anterior horn cells in the spinal cord was manifested by muscle atrophy and pathological changes consistent with nerve fiber denervation and reinnervation, and it was proposed that HERV-Kenv protein may contribute to neurodegeneration [66].

A series of licensed anti-HIV inhibitors were tested in cultured cells against HERV-K expression; *in vitro* HERV-K was inhibited by inhibitors of reverse transcriptase and integrase, while protease inhibitors were not as effective in inhibiting HERV-K virus as HIV [67].

*In vivo* HERV-K activation and response to antiretroviral therapy was monitored in the plasma of five HIV+ patients with motor neuron disease. Three patients had reversal of symptoms within 6 months; they had elevated HERV-K levels that responded to optimization of antiretroviral therapy for CNS penetration. The conclusion was that monitoring of HERV-K levels may help to guide treatment [68].

Based on HERV-K sensitivity to anti-HIV drugs *in vitro* and *in vivo* [67,68], two phase 1 clinical trials on HIV-negative ALS patients treated with anti-HIV drugs are ongoing in Australia and UK [69], and in USA [70]. So far, no results have been released by the latter study, while the former reported that the therapy was safe and well tolerated, and that there was suggestive indication for a possible biological response in some pharmacodynamic and clinical biomarkers [69].

Data partly conflicting with the above findings of HERV-K expression in ALS patients were also reported. Twenty-four different transcribed HML-2 loci were identified in brain and spinal cord tissue samples from ALS patients and controls [71], by generating and mapping HML-2-specific cDNA sequences, without significant differences between ALS and controls, and opened up the possibility that HML-2 proteins other than canonical full-length Env may have to be considered when studying the role of HML-2 in ALS disease [71]. In another study, HERV-K *gag*, *pol* and *env* transcripts were evaluated in total RNA from *postmortem* premotor cortex of ALS patients and controls; geometric mean HERV-K RNA expression levels were not found to be different between patients and controls, and raised doubts about HERV-K role in ALS pathogenesis [72].

In evaluating data on HERVs, it must be pointed out that some caveat must be considered, due to the peculiarity of the study target. At variance with cellular genes, generally present in single copy/haploid genome, each HERV family is composed by multiple elements, largely similar each other in sequences, but slightly or largely different, for a wide variety of mutations and coding capacity, which may be dysregulated by epigenetic changes, inflammatory and environmental triggers. In this scenario, it is highly possible that “*the*” relevant HERV element is expressed at relatively low levels and masked by the bulk of expression, largely dysfunctional, of other elements of the same family. This implies that it is mandatory first to identify the relevant, functional, retroelement, and to create and validate selective assays. Functional transcripts can be enriched by processing polyA+ RNA, instead of total RNA, as done in almost all the studies of this Section, except ours, and to search also for the final protein, recognized by a specific antibody. Another major issue is the choice of the controls: since HERV-K has oncogenic potential and is activated in cancer, and these patients should not be considered equivalent to healthy controls. In this respect, no correlation of HERV-K levels with age or gender of non-ALS donors was found in total RNA of brain samples, [71,72]. Conversely, in plasma from healthy volunteers, as shown on Figure 4, we observed that the levels of HERV-K(HML-2)*env* polyA+ RNA increase significantly with the age of the donors.

### 3.3. HERV-K as Contributor to ALS Pathogenesis

An unquestionable issue is that transgenic mice expressing the HERV-K*env* gene developed an ALS-like motor neuron dysfunction [65]. Thus, the expression of the HERV-K*env* gene alone was able to cause the motor neuron disease. Moreover, cultured human neuronal cells transfected with the HERV-K*env* gene (or with the whole HERV-K genome) showed retraction of neurites and decrease in cell numbers, suggesting that the env protein contributed to neurotoxicity and neuronal death [65]. A similar pattern of neurotoxicity was also achieved by activation of the endogenous HERV-K through the LTRs [65].

It must be reiterated that TDP-43, deposits of which are thought to be critical in motor neuron degeneration and considered the final hallmark of ALS, was discovered after its capacity to bind the TAR domain of the HIV retrovirus [73]. The HIV TAR (trans-activation response) element is an RNA domain of the viral genome, required for trans-activation of the HIV LTR promoter and for viral replication. It has a stem-loop structure which serves as binding site for both the HIVtat protein and cellular factors, and this interaction stimulates the activity of the LTR promoter and viral gene expression.

TDP-43 is an aggregation-prone member of the family of DNA-RNA-binding proteins, important negative regulators of alternative splicing, physiologically involved in several aspects of RNA metabolism, stress granules and aggregation, mitochondrial function and autophagy regulation [74,75].

In the Drosophila model of ALS, the expression of human TDP-43 in neurons and glia impaired the silencing of small interfering RNA, and activated the gypsy endogenous retrovirus, which, in turn, caused the degenerative phenotypes of the flies [76].

In postmortem brain tissues from ALS patients, the HERV-K reverse transcriptase protein co-localyzed with the TDP-43 protein in cortex motor neurons [65]. As stated previously, expression of HERV-K or its env protein in human neurons was neurotoxic [66]. TDP-43 was shown to upregulate HERV-K expression in human neurons and cell lines, at both the RNA and protein levels. The binding sites for TDP-43 were identified in the LTR region of HERV-K, and were different from that for HIVtat (the effect of which is additive to that of TDP-43 in HERV-K activation), suggesting that they act on different sites [66]. Thus, HERV-K expression within neurons of patients with ALS may contribute to neurodegeneration and disease pathogenesis [66].

Through *in vitro* experiments on astrocytic and neuronal cells and analysis of ALS brain tissues, Manghera and colleagues found that TDP-43 facilitates HERV-K reverse transcriptase protein deposition in human neurons, and that human astrocytes and neurons have cell-type specific differences in their ability to express and clear HERV-K reverse transcriptase proteins during inflammation and proteasome inhibition; astrocytes, but not neurons, cleared excess HERV-K proteins through stress granule formation and autophagy [77]. They proposed HERV-K “proteinopathy” as a novel mechanism of neuronal damage in ALS disease.

To give insights on the HERV-K effects, Ibba and colleagues disrupted the HERV-K(HML-2)*env* gene by CRISPR/SaCas9 endonuclease technology, in cultured cells normally expressing HERV-Kenv mRNA and proteins [55]. When the HERV-K*env* gene was disrupted, there was a down-modulation of important regulators of cell expression and proliferation, involved in signaling, RNA-binding and alternative splicing, as EGF-R, NF-κB, SF2/ASF and TDP-43. On Figure 5, it is shown that the disruption of the HERV-K*env* gene, is followed by the reduction, not only of the levels of HERV-K*env* mRNA, but also of those of TDP-43. Hence, the disappearance of the HERV-Kenv protein, accompanied by a substantial reduction of the TDP-43 protein [55]. TDP-43 is critical for cell survival, and optimal levels of the TDP-43 protein are maintained, by an autoregulatory pathway; levels of TDP-43 (either wild-type or mutant protein) increased by less than a factor of 2 are highly deleterious, and lead to neurodegeneration [44]. Our data show that not only TDP-43 activates HERV-K, as reported above [65,66,77], but also that HERV-K is responsible for TDP-43 activation, indicating the existence of a vicious circle of reciprocal HERV-K/TDP-43 activation and of their pathogenic potential [55].

## 4. HERVs and Neuroaids

Around 50% of HIV-positive adults have some neurological impairment [78] and neuroAIDS is a well-recognized syndrome [18]. Multiple pathogenic mechanisms contribute to HIV-related neurodegeneration; however, HIV replication accounts for a minor part of neurodegeneration, as in the brain HIV replicates mainly in perivascular macrophages and microglia, while it is generally restricted in astrocytes and neurons, and the reactivation of bystander viruses, included that of HERVs, contributes significantly to neuropathogenesis [44].

*In vivo*, HERV-K activation in HIV+ patients has been reported in brain and blood [79,80,81], and HERV-K testing has been proposed as surrogate biomarker of HIV disease and therapy outcome [80].

In an analysis of fours HERV families in the brain of neuroAIDS patients (HERV-W, HERV-K, HERV-E, and HERV-H) [82], a significant upregulation of HERV-W and HERV-K RNA was detected in brain tissue from patients with AIDS dementia, the expression of HERV-H was slightly increased, while that of HERV-E was significantly reduced. The changes in HERV expression were associated with inflammation, monocyte differentiation and macrophage activation, and showed differential modulation of the various HERVs [82].

In the blood of HIV+ patients, the expression of MSRV/HERV-Wenv was only recently studied [83], in comparison to healthy and MS persons. MSRV/HERV-Wenv expression was slightly, but not significantly, lower in HIV patients with respect to healthy controls. They found that T cells are HERV-W/MSRV-negative, confirming previous reports [43,45]. Of the nine HIV+ patients, only three were naïve and six were under antiretroviral therapy; this is in line with the *in vitro* sensitivity of MSRV/HERV-W to the antiretroviral drugs received *in vivo* by the same patients. The values of the MS group were significantly higher than controls, and confirmed previous reports [27,43,45].

The mechanism of HERV activation by HIV was shown to be mediated by HIVtat indirectly, through tat binding to the Toll-Like Receptor-4, and early induction of tumor necrosis factor-α, which is the final responsible of Tat activation of HERVs [44]. Tat effects are counteracted by binding of the epidermal growth factor to its receptor, but its actual role in the brain remains to be elucidated [84].

Similarities between ALS and neuroAIDS have been underlined [85]. An ALS-like syndrome occurs in 3.5 per 1000 HIV-infected persons, which regressed under therapy with anti-HIV drugs [57,64]. These patients showed expression of HERV-K in blood [86], the levels of which fell in all patients after therapy with antiretroviral drugs [68]. Independent groups have shown that HERV-K proteins accumulate in cortical neurons of patients with HIV infection [87]. Local neuroinflammation is likely to be a key driver of HERV-K expression in the brain, as supported by a study of ALS neuropathology [77].

Another important similarity between ALS and neuroAIDS is the formation of neurotoxic nuclear and cytoplasmic TDP-43 deposits in neurons [85], and it was posited that, if sub-optimal dosages of antiretroviral therapy occur in HIV+ patients in the CNS, then HIV-enhanced HERV-K in the brain might stimulate locally the expression of TDP-43, as is was shown *in vitro* [55]. This opens the possibility that therapeutics overcoming TDP-43-mediated pathology in ALS could be clinically useful in HIV-associated neurological disease [85].

## 5. Conclusions

HERVs are ancestral genetic parasites, which constitute around one tenth of our genome, with some coding capability. Sometimes, the host has the advantage of some of them, for example, as HERV-W/Syncytin-1, for embryo implantation on placenta. The expression of the few intact HERV genes depends on several factors.

When the coding potential is retained, the expression of different HERVs varies, even within the same individual, depending on several physiological factors, and their baseline predisposition to be expressed may be regulated epigenetically by different micro- and macro-environmental triggers, such as inflammation, viral and microbial infections, etc.

It is no wonder that the increased expression of one or more HERVs has been detected in various physiological or pathological conditions, in one or more body sites. Various clinical situations have been attributed to one or more HERVs, particularly with respect to some neurological diseases and cancer, and its detection has been proposed as a biomarker, to monitor individual disease behavior and therapy outcome.

The key problem is to differentiate the expression of a HERV as cause or effect of a disease. For most of the HERVs, a bystander role seems likely. A major caveat is that each HERV family is composed od multiple elements, largely similar each other in sequences, but slightly or largely different, for a wide variety of mutations and coding capacity, which may be dysregulated by epigenetic changes, inflammatory and environmental triggers, not necessarily in the same sense. An example is the following: two *env* genes in the same HERV-W family, HERV-W/MSRV*env* and HERV-W/Syncytin-1, differ only for for presence/absence, in the 1629 nucleotides of the gene, of twelve nucleotides in the intracellular domain, or four aminoacids in the env protein. In circulating monocytes, HIVtat stimulates HERV-W/MSRV*env* by 3-4-folds, while heavily reduces HERV-W/Syncytin-1: this totally opposite behaviour can be detected only by using discriminatory primers, while the inhibition of HERV-W/Syncytin-1 is masked by the stimulation of the other *env* gene, if primers recognizing the *env* genes of the whole HERV-W family are used [44].

In this scenario, it is highly possible that “*the*” relevant HERV element is expressed at low levels and masked by the bulk of expression, largely dysfunctional, of other elements of the same family. This may explain discordant findings. Therefore, it is mandatory to first identify the relevant, functional, retroelement, to be correlated with the disease under study, by specific and validated assays.

To be used as a biomarker, a statistically significant correlation between the expression of a certain HERV and the disease onset and/or behavior must be found.

As for a pathogenic role of an HERV, there are criteria defining causal connections between a HERV and a disease; they include the development of animal models, and disease modulation in humans by anti-HERV therapeutic antibody.

So far, a statistically significant correlation between HERVs and diseases was achieved for HERV-W and multiple sclerosis. The demonstration of a true pathogenic potential (disease reproduction in transgenic animals) has been achieved for HERV-W and multiple sclerosis, and for HERV-K and amyotrophic lateral sclerosis. For both diseases, clinical trials, against HERV-W and HERV-K respectively, are in progress.

Intriguing HERV-related similarities between ALS and neuroAIDS in the neurodegeneration process have been observed, which could lead eventually to innovative clinical applications.

Among outstanding unsolved questions is the comprehension of the interactions between HERVs and the human genome where they are inserted; these interactions may vary depending on polymorphic human genes with pathogenic potential, but also with respect to HERV polymorphisms between human populations and individuals and non-ubiquitous HERV insertions, which, being present in small clusters of humans, may go unappreciated, since absent in current genome databases. Databases of these clusters, genome-wide sequencing and transcriptomics will help to clarify these issues. It is relevant also to understand the dynamics of the possible effects of a HERV, which may start long before the onset of a disease. Lastly, there is the urgent need to standardize the HERV nomenclature and to provide retroelement-specific validated assays. The above points may reinforce the link of some complex diseases to a HERV.

## Figures and Tables

**Figure 1 ijms-20-03706-f001:**
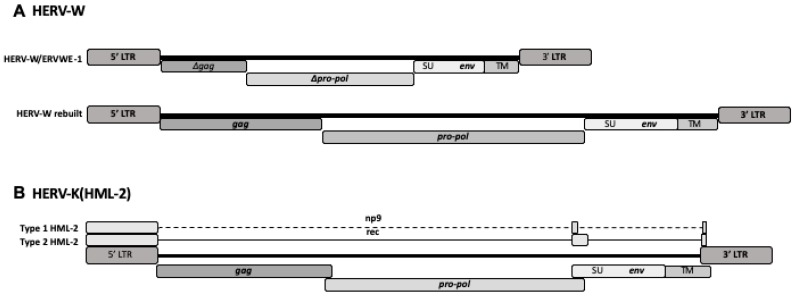
Schematic structure of the human endogenous retrovirus (HERV)-W and HERV-K(HML-2) proviruses in human DNA. Typically, complete proviruses of both families have at the 5′ and 3′ ends two long terminal repeats (LTRs), which allow and regulate the expression of viral genes, if preserved. Major genes are *gag*, encoding the capsid proteins; *pro-pol*, encoding viral protease and reverse transcriptase; *env*, encoding the envelope proteins. The wide majority of the members of both families are defective. (**A**) The top provirus represents the HERV-W/ERVWE-1 element, located on human chromosome 7q21.22, which retains intact only the *env* open reading frame, coding for the functional HERV-W/Syncytin-1 protein. The bottom diagram represents the reconstructed HERV-W complete provirus. HERV-W gag, pol, and env proteins from non-localized HERV-W elements have been detected repeatedly [2,3]. The HERV-W family has at least 300 members, not including the solo LTRs [7]. (**B**) The HERV-K(HML-2) subgroup of the HERV-K family. This subgroup has around 60 members with coding capability [8], which are divided in types 1 and 2, by the presence, only in HERV-K(HML-2) type 1, of a 292-bp deletion at the *pol-env* boundary, which eliminates a splicing site and, instead of Rec, creates the Np9 protein, a pivotal switch of several signaling pathways, which has oncogenic properties. The diagrams are not drawn to scale.

**Figure 2 ijms-20-03706-f002:**
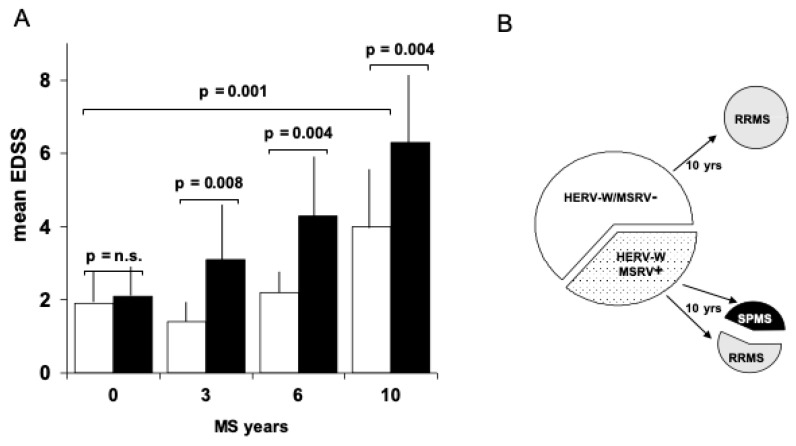
Clinical evolution along ten years of 22 multiple sclerosis (MS) patients differing at onset only for presence/absence of HERV-W/MSRV in the cerebrospinal fluid (CSF). (**A**) Mean expanded disability status scale (EDSS). Over the 10 years, the patients with HERV-W/MSRV(+) CSF at the onset (black bars) progressed with significantly higher disability, with respect to the patients with HERV-W/MSRV(-) CSF (white bars). (**B**) MS forms of the cohort at entry and after 10 years of follow-up. Forty-three % of the patients with HERV-W/MSRV(+) CSF converted into secondary-progressive MS (SPMS), while those with HERV-W/MSRV(-) CSF at onset remained in relapsing-remitting MS (RRMS) [29,30,31].

**Figure 3 ijms-20-03706-f003:**
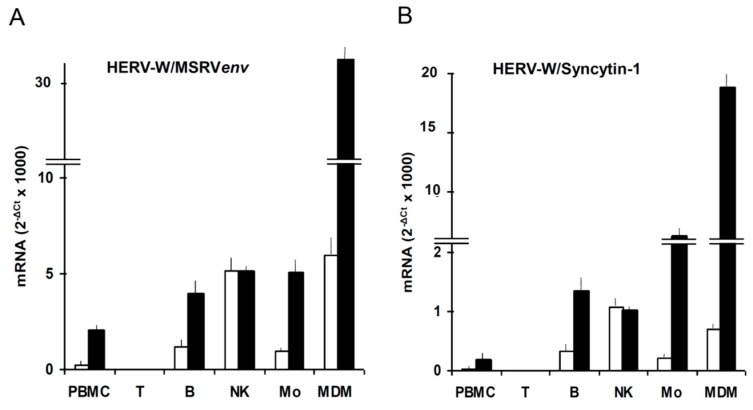
Expression of HERV-W/MSRVenv and HERV-W/Syncytin-1 mRNAs by PBMC subsets, as such and after exposure to the EBVgp350 protein. PBMC from HERV-W/MSRV(+) donors were tested as such, and after immunobeads separation in CD3+T, CD19+ B, CD56+/CD192/CD32 NK and CD192/CD3–/CD562 monocyte subsets; monocyte aliquots were also differentiated into macrophages (MDM). The cells were treated overnight with recombinant EBVgp350 protein, then were harvested and processed for mRNA extraction, reverse transcription and real time PCR with primers selective for HERV-W/MSRVenv and HERV-W/Syncytin-1, as published [43]. (**A**) HERV-W/MSRVenv. (**B**) HERV-W/Syncytin-1. Open bars: basal levels; black bars: mRNA levels of cells exposed *in vitro* to EBVgp350.

**Figure 4 ijms-20-03706-f004:**
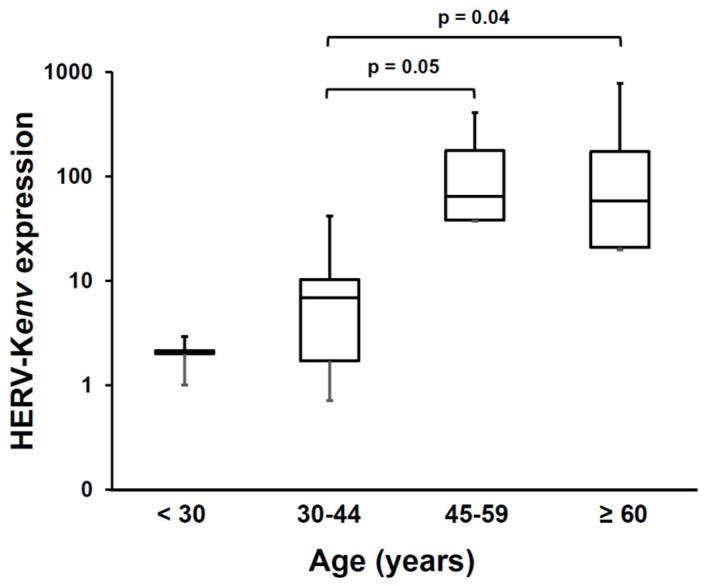
Age-related levels of HERV-K(HML-2)*env* polyA+ RNA in plasma from healthy volunteers. Published methods were used for blood fractionation and mRNA extraction [27,44], and for HERV-K(HML-2)*env* retro-transcription and real time amplifications [55]. The data were calculated by the 2^-Ct^ method, and are expressed in arbitrary units. The absence of cellular contaminants was assured by evaluation of glyceraldehyde-3-phosphate dehydrogenase housekeeping gene transcripts, before and after the reverse transcription step.

**Figure 5 ijms-20-03706-f005:**
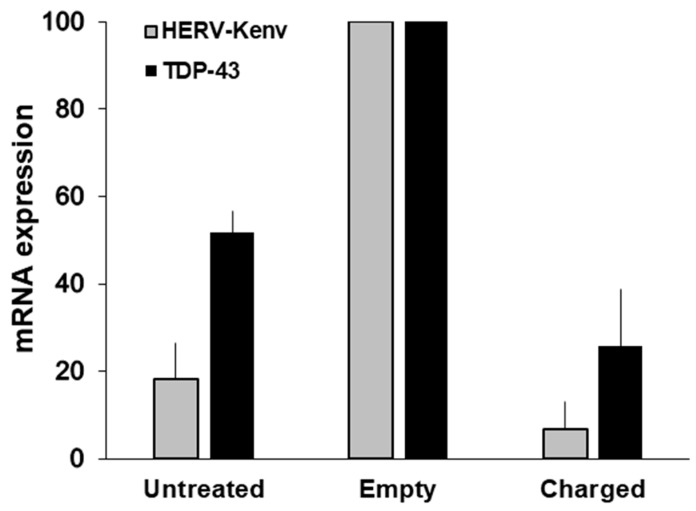
Levels of HERV-K*env* and TDP-43 mRNAs in LNCaP cells transfected either with the empty plasmid or with charged plasmid. The histograms represent the mean levels of HERV-K*env* and TDP-43 mRNAs, as detected by RT-qPCR, and are reported as percent of the value of mock-transfected cells in each experiment. Experimental conditions: Untreated cells, and cells either transfected with the empty plasmid, or with the plasmid charged with the Km3gRNA. For controls of specificity, methods and details, see [55].

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
