# Peer review of "Expression of HERV Genes as Possible Biomarker and Target in Neurodegenerative Diseases"

_ijms, 2019, doi:10.3390/ijms20153706_

Reviewer 1 Report

In the review article titled ‘Expression of HERV genes as possible biomarker and target in neurodegenerative diseases’, the authors have very succinctly summarized the literature on HERV and their correlation with neurodegenerative diseases. However, I have few concerns, addressing them would improve the quality of the review. The article under current format can be considered article for publication in IJMS if authors address the concerns raised below.

Some of the concerns

1)      Although, the figures/plots used in the review are picked up from the previously published articles of the authors, it might have copyright issues. Please clarify on that to the editors of IJMS.

2)      Overall the review is very well-written. But the authors fail to conclude the same. They have reiterated the same sentences picked from elsewhere in the article. I think it would be a good idea to include outstanding questions in the field and how to approach those questions.

Author Response

Response to Reviewer 1 Comments

Overall the review is very well-written:

We thank this Reviewer.

The changes in the revised version are highlighted in yellow.

Point 1: Although, the figures/plots used in the review are picked up from the previously published articles of the authors, it might have copyright issues. Please clarify on that to the editors of IJMS.

Response 1: OK, we are aware of copyrights, and all the Figures have been drawn ex novo.

Point 2:  Overall the review is very well-written. But the authors fail to conclude the same. They have reiterated the same sentences picked from elsewhere in the article. I think it would be a good idea to include outstanding questions in the field and how to approach those questions.

Response 2: OK, a new paragraph is included in the revised paper, on page 12, lines 475-474, as follows:

“Among outstanding unsolved questions is the comprehension of the interactions between HERVs and the human genome where they are inserted; these interactions may vary depending on polymorphic human genes with pathogenic potential, but also with respect to HERV polymorphisms between human populations and individuals and non-ubiquitous HERV insertions, which, being present in small clusters of humans, may go unappreciated, since absent in current genome databases. Databases of these clusters, genome-wide sequencing and transcriptomics will help clarifying these issues. It is relevant also to understand the dynamics of the possible effects of a HERV, which may start long before the onset of a disease. Lastly, there is the urgent need to uniform HERV nomenclature and to provide retroelement-specific validated assays. The above points may reinforce the link of some complex diseases to a HERV.”

Reviewer 2 Report

This manuscript is the review paper that the expression of HERV W and K as possible biomarker and target in neurodegenerative diseases.

This review paper is so interesting. However, there are some major points and minor points that the authors may consider.

Major points:

Authors explained the correlation between HERV W/K and MS/ALS in this paper. However, the diagram of genomic structure of these two viruses did not indicate. If authors can provide the differences in the genomic structure of HERV-W and HERV-K in new figure or table, it will be helpful for the reader to understand.

Although authors mentioned neurodegenerative diseases in title, they referred only three neurodegenerative diseases including MS, ALS, and neuroAIDS. I should recommend to add the HERV data for prion disease and schizophrenia.

Authors suggested the possibility of HERV as a biomarker, however authors did not offer sensitivity and specificity of HERV in diagnosis of MS or ALS. You should offer this information.

Minor points:

Abbreviations should be checked.

For examples,

 Line 107: amyotrophic lateral sclerosis (ALS)

 Line 238: amyotrophic lateral sclerosis (ALS, see below)

 Line 250: amyotrophic lateral sclerosis

 Line 251: Amyotrophic lateral sclerosis (ALS)

 Line 455: amyotrophic lateral sclerosis

Some errors should be checked.

For examples,

Line 235: lateral amyotrophic sclerosis à amyotrophic lateral sclerosis

Line 390: Figure 3 à  Figure 4

Author Response

This review paper is so interesting.:

We thank this Reviewer.

The changes in the revised version are highlighted in yellow.

Major points:

Point 1:  Authors explained the correlation between HERV W/K and MS/ALS in this paper. However, the diagram of genomic structure of these two viruses did not indicate. If authors can provide the differences in the genomic structure of HERV-W and HERV-K in new figure or table, it will be helpful for the reader to understand.

Response 1: OK,   a new Figure 1 has been added, on Page 3, lines 108-123 of the revised version.

Point 2:   Although authors mentioned neurodegenerative diseases in title, they referred only three neurodegenerative diseases including MS, ALS, and neuroAIDS. I should recommend to add the HERV data for prion disease and schizophrenia.

Response 2: Sorry, As specified in the old version of the paper, on page 2, lines  77-82, and on page 3, lines 97-103, the detection of a HERV does not means necessarily association nor causation, and there are criteria to define the correlation between a HERV and a disease.  As stated on page 3, lines 104-107, we reported only the data, confirmed by independent groups, getting close to the above criteria, and, so far, this is not the case for prions nor for schizophrenia.

Point 3:   Authors suggested the possibility of HERV as a biomarker, however authors did not offer sensitivity and specificity of HERV in diagnosis of MS or ALS. You should offer this information.

Response 3:  OK.  The problem of differential evaluation of individual members of the same family, was addressed for HERV-W, in the old version of the paper, on the footnote 5, on page 11, now on page 12 of the revised paper. Discriminatory assays for HERV-W elements have been created and validated only by our group, and published in the paper by Mameli et al., 2009, reference #25), and used (confirming independently our findings), by Garcia-Montojo et al., 2013 (ref.# 36). All the other studies used primers recognizing the env of the whole HERV-W family (or other HERV-W genes). At the protein level only antibody against the env of the whole HERV-W family are available. The sensitivity and specificity of our discriminatory PCR assays were published in the paper by Mameli et al., 2009, reference #25. A sentence was added on page 4, lines 130-131 of the revised paper (“…, by selective primers, whose specificity, sensitivity and validation has been reported [25]”).  Another sentence has been changed on page 8, lines 331-332 of the revised paper (“… first to identify the relevant, functional, retroelement, and to create and validate selective assays”). As for the HERV-K (HML-2) evaluation in relationship to ALS, there are very few reports, and we used the same primers used in the first paper in the field (Douville et al., 2011, ref. # 64 of the old version, now ref. #65 of the revised version), and others deriving from scientists of the first study. No data are available on sensitivity and specificity of HERV-K PCR. We specified it better on page 3, lines 99-110: “…must be found, by specific and validated assays.”, and on page 12, lines 461-462 of the revised paper: “….to be correlated with the disease under study, by specific and validated assays.”.

Minor points:  

Abbreviations should be checked. Some errors should be checked.

Response 4:  Ok, it has been done.

Reviewer 3 Report

In this study, the authors summarized the current findings on the expression of the Human endogenous retroviruses (HERVs) in neurodegenerative human diseases. They described that there is a statistically significant correlation between HERVs expression with multiple sclerosis and amyotrophic lateral sclerosis. However, the authors conclude that also if the HERVs could be proposed as biomarkers to monitor individual disease, the problem could be the differentiation of the expression of HERVs as cause or effect of a disease. Consequently, they suggest to identify the functional retroelement, to be correlated with the disease under study. Moreover, they summerized that a statistically significant correlation between HERVs and diseases was achieved for multiple sclerosis and amyotrophic lateral sclerosis, allowing for both diseases clinical trials, ongoing, against the specific HERVs.

Comments:

1. Line 38: add a reference

2. Line 58: delete for

3. Line 123: correct with “for identifying” with to identify

4. Line 142: delete had (because is duplicated)

5. Line 231: delete for

6. Line 267: add a bracket after “ALS cases”

7. Line 281: this sentence is a repetition of line 278, I would suggest to delete the last sentence and add in the first one the reference to the protein, in order to combine both sentences.

8. Line 454: delete was

9. I would suggest to mention also something reported in this reference: Göttle P, Förster M, Gruchot J, Kremer D, Hartung HP, Perron H, Küry P. Rescuing the negative impact of human endogenous retrovirus envelope protein on oligodendroglial differentiation and myelination. Glia. 2019 Jan;67(1):160-170. doi: 10.1002/glia.23535. Epub 2018 Nov 14. PubMed PMID: 30430656.

Author Response

The changes in the revised version are highlighted in yellow.

9 minor Comments:

1. Line 38: add a reference. 2. Line 58: delete for. 3. Line 123: correct with “for identifying” with to identify. 4. Line 142: delete had (because is duplicated). 5. Line 231: delete for. 6. Line 267: add a bracket after “ALS cases”. 7. Line 281: this sentence is a repetition of line 278, I would suggest to delete the last sentence and add in the first one the reference to the protein, in order to combine both sentences.8. Line 454: delete was. 9. I would suggest to mention also something reported in this reference: Göttle P, Förster M, Gruchot J, Kremer D, Hartung HP, Perron H, Küry P. Rescuing the negative impact of human endogenous retrovirus envelope protein on oligodendroglial differentiation and myelination. Glia. 2019 Jan;67(1):160-170. doi: 10.1002/glia.23535. Epub 2018 Nov 14. PubMed PMID: 30430656.

Response 1: OK,  We  made the small corrections indicated, including Minor Point 1 (two References were added at the line 38, page 1),  and Minor Point 4 (the Past Perfect “had had” has been changed in Past Tense “had”, as requested by this Reviewer. We added the suggested new reference to the paper by Göttle et al., 2019, as the new reference #51. Accordingly, on page 7, lines 248-249 of the revised paper, the sentence has been changed: “….protein, via myeloid cells, directly harms axons, and that the damage can be overcome by anti-HERV-Wenv antibody [50-51]”.

Round  2

Reviewer 1 Report

The revised version of the manuscript looks good and succinct.

The response of authors for the concern I have raised previously about the figures, especially figure 5 is not satisfactory.

Making it anew is not solution for it. If I understand it correctly, the blot is remade, essentially using the same images by cropping, which still might pose copyright and image duplication threats. Please look into it and discuss this matter with editor.

If the above concerns are addressed, I recommend the article for publication in IJMS.

Author Response

Making it a new is not solution for it. If I understand it correctly, the blot is remade, essentially using the same images by cropping, which still might pose copyright and image duplication threats.

Response : OK: in the new Figure 5, the Panel (b) has been deleted, and the previous Figure 5a was drawn ex novo, using values from experiments different from those used for the previous version. The new Figure 5 is located now at the end of the related paragraph, on page 10, lines 390-395.

Reviewer 2 Report

The revised manuscript by Dolei A. et al. was markedly improved and satisfactorily modified for reviewer comment. Thus, I am recommending acceptance for publication in International Journal of Molecular Science.

Author Response

Thanks